# Physicochemical Stability of Nab-Paclitaxel (Pazenir) Infusion Dispersions in Original Glass Vials and EVA Infusion Bags

**DOI:** 10.3390/pharmaceutics16111372

**Published:** 2024-10-26

**Authors:** Helen Linxweiler, Judith Thiesen, Irene Krämer

**Affiliations:** Department of Pharmacy, University Medical Centre of Johannes Gutenberg-University, Langenbeckstraße 1, 55131 Mainz, Germany; judith.thiesen@unimedizin-mainz.de (J.T.); prof.dr.irene.kraemer@unimedizin-mainz.de (I.K.)

**Keywords:** nab-paclitaxel, nanoparticles, in-use stability, parenteral preparation

## Abstract

Background/Objectives: The study objective was to determine the physicochemical stability of nab-paclitaxel (Pazenir) ready-to-use (RTU) dispersion for infusion in original glass vials and ready-to-administer (RTA) infusion dispersion in EVA infusion bags. Methods: Triplicate test dispersions were prepared and stored light protected for a maximum of 28 days either in the original glass vials (RTU) at 2–8 °C or in EVA infusion bags (RTA) at 2–8 °C and at 25 °C. Directly after reconstitution and on days 1, 3, 5, 7, 14, 21, and 28 samples were withdrawn and paclitaxel concentrations assayed by a stability-indicating HPLC method. In parallel, pH and osmolality were measured. In a second series, test dispersions were stored over a 14-day period and inspected daily for visible particles and colour changes. Samples were taken daily for particle size analysis. Integrity and particle size distribution of the nanoparticles were determined by dynamic light scattering (DLS) and albumin monomers, dimers, oligomers, or polymers by size-exclusion-chromatography (SEC). Results: Non-redispersible particles were observed in test dispersions on day 5 (RTA 25 °C), day 7 (RTA 2–8 °C), and day 11 (RTU 2–8 °C). DLS analysis revealed out-of-specification results for the polydispersity index from day 7 (RTA 25 °C) and day 12 (RTU, RTA refrigerated). Paclitaxel concentrations remained >95% of the initial concentrations for 7 days (RTU 2–8 °C, RTA 25 °C) and for 14 days (RTA 2–8 °C). All test dispersions met the specifications regarding the oligomeric status of albumin, pH, and osmolality over the investigation periods. Conclusions: Stability of nab-paclitaxel dispersions is limited by the release of water-insoluble paclitaxel from the nanoparticles and subsequent crystallisation and by formation of insoluble albumin aggregates. Based on our overall results, shelf life of refrigerated RTU and RTA nab-paclitaxel dispersions is limited to 7 days. Shelf life of RTA nab-paclitaxel dispersions stored at room temperature is limited to 4 days. Careful visual inspection of nab-paclitaxel dispersions after reconstitution and prior to administration is highly recommended to detect non-redispersible particles.

## 1. Introduction

Nab-paclitaxel represents a medicinal product belonging to the group of non-biological complex drugs (NBCDs). It is a solubilizer-free, lyophilised albumin-bound formulation of paclitaxel, which is used in anticancer therapy. The originator was licensed in the EU in 2008 (Abraxane^®^ by Abraxis BioScience, Los Angeles, CA, USA) and since 2019 generic Pazenir 5 mg/mL powder for dispersion for infusion (ratiopharm GmbH) is marketed.

The nab-technology (nab = nanoparticle albumin-bound) enables encapsulation of lipophilic active substances into nanoparticles. Paclitaxel is a suitable drug candidate for the nab-technology, as the traditional formulation contains Cremophor EL and ethanol as solubilizer. Cremophor-based paclitaxel can cause neurological toxicity, neutropenia, and acute hypersensitivity reactions [1]. Thus, it needs prolonged infusion time and premedication with corticosteroids, antihistamines, and H2-receptor antagonists. In contrast, nab-paclitaxel can be infused in only 30 min and is not associated with hypersensitivity reactions, making premedication unnecessary. Given these advantages, nab-paclitaxel has emerged as a significant alternative to traditional paclitaxel formulations.

To formulate nab-paclitaxel, the water-insoluble drug paclitaxel is mixed with human serum albumin (HSA) in an aqueous solvent and homogenised under high pressure. Thereby, nab-paclitaxel nanoparticles (100 to 200 nm in size) are formed [2]. The nanoparticles consist of paclitaxel-bound albumin complexes with several paclitaxel molecules bound non-covalently to the lipophilic pockets on the albumin molecule surface [3]. The albumin concentration in the nanoparticle dispersion is 3–4%, which is equivalent to the albumin concentration in human blood [4]. After intravenous infusion, the nanoparticles dissociate rapidly into smaller albumin-bound complexes of about 10 nm in size (see Figure 1) [4]. These complexes accumulate in tumour tissue through Enhanced Permeation and Retention (EPR) mechanism. Furthermore, albumin enhances transport of paclitaxel across endothelial cells mediated by a 60 kDa glycoprotein (gp60) receptor on the endothelial cell surface and accumulates in the tumour area by binding to the extracellular protein Secreted Protein Acidic Rich in Cysteine (SPARC) [5].

Finished Pazenir medicinal product contains lyophilised nab-paclitaxel 5 mg/mL, sodium caprylate and N-acetyl-DL-tryptophan (albumin stabilisers), sodium chloride, hydrochloric acid, and sodium hydroxide as excipients. The lyophilisate is to be reconstituted with 0.9% sodium chloride and transferred into an empty infusion bag prior to administration. The nab-paclitaxel dose range of 100–260 mg/m^2^ body surface area results in an infusion volume of about 30–100 mL [6]. According to the summary of product characteristics (SmPC), the reconstituted dispersion is physicochemically stable in the vial and infusion bag for 24 h at 2–8 °C followed by 4 h at 15–25 °C. Additional data on the physicochemical stability of ready-to-use (RTU) nab-paclitaxel (Pazenir) dispersions in the original vial are published [7]. The authors report that RTU nab-paclitaxel 5 mg/mL dispersions were stable for 22 days when stored refrigerated or for 7 days when stored at room temperature. To our best knowledge, prolonged stability data of ready-to-administer (RTA) nab-paclitaxel dispersions (originator, generic) are not published yet.

The objective of the study was to determine the physicochemical in-use stability of reconstituted Pazenir lyophilised powder in the original glass vial (RTU) and in ethylene vinyl acetate (EVA) infusion bags (RTA). Physicochemical instability of reconstituted and RTA nab-Paclitaxel is related to disintegration or aggregation of the nanoparticles [4], precipitation of paclitaxel [8,9,10], and degradation reactions of albumin [11]. In-vitro disintegration of the nanoparticles leads to smaller albumin-bound paclitaxel complexes and free insoluble paclitaxel [4]. Physicochemical stability of nab-paclitaxel dispersions was investigated by dynamic light scattering (DLS), size exclusion chromatography (SEC), and visual inspection. Chemical stability was examined via reversed-phase high-performance liquid chromatography (RP-HPLC), osmolality, and pH measurement. Results of the in-use stability study will contribute to more efficient preparation and storage practices of nab-paclitaxel infusion dispersions in healthcare settings.

## 2. Materials and Methods

### 2.1. Stability Criteria

Physicochemical stability of nab-paclitaxel dispersions was determined by analytical methods and specifications given in Table 1.

### 2.2. Chemicals and Reagents

Pazenir 5 mg/mL powder for dispersion for infusion (ratiopharm GmbH, Ulm, Germany, batches 22A07M, 22A16K, 22B20L); Paclitaxel chemical reference substance CRS (EDQM, Strasbourg, France, catalogue no. Y0000698, batch 4), human albumin CRS (EDQM, Strasbourg, France, catalogue no. Y0001913, batch 1.0); 0.9% NaCl infusion solution, 500 mL freeflex^®^ bag (Fresenius Kabi, Bad Homburg, Germany, batch 14QD7330); MIB^®^ 150 mL EVA infusion bags (material ethylene vinyl acetate, Hemedis GmbH, Weißenborn, Germany, batch CGH70201).

Paclitaxel RP-HPLC assay: mobile phase: acetonitrile HPLC grade (Honeywell International, Morristown, USA, batches L3320, M0830), water HPLC grade (AppliChem GmbH, Darmstadt, Germany, batches 2Q011535, 2K012127). Albumin SEC: mobile phase sodium dihydrogen phosphate dihydrate (Merck KGaA, Darmstadt, Germany, batch K95814542244), di-sodium hydrogen phosphate dihydrate (Merck KGaA, Darmstadt, Germany, batch K51032380908), sodium chloride (Euro OTC Pharma GmbH, Bönen, Germany, batch 2108012-02), water HPLC grade (AppliChem GmbH, Darmstadt, Germany, batches 2Y012453, 2K012127).

### 2.3. Preparation of Test Dispersions

#### 2.3.1. Reconstituted Nab-Paclitaxel RTU Test Dispersions in Original Glass Vials

For each test series, three vials of Pazenir 5 mg/mL (100 mg) powder for dispersion for infusion (three different batches) were reconstituted with 20 mL 0.9% NaCl infusion solution each, slowly swirled and inverted to achieve a homogenous dispersion. Test dispersions were stored at 2–8 °C protected from light in the refrigerator for a maximum period of 28 days (series 1) and 14 days (series 2).

#### 2.3.2. Reconstituted Nab-Paclitaxel RTA Test Dispersions in EVA

For each test series, nine vials of Pazenir 5 mg/mL (100 mg) powder for dispersion of three batches were reconstituted with 20 mL 0.9% NaCl infusion solution each, withdrawn from the vials and transferred into three empty EVA infusion bags (300 mg nab-paclitaxel per infusion bag). Test dispersions were stored light-protected either at 2–8 °C in a refrigerator or at 25 °C in a climate chamber (Memmert ICH260L, Memmert, Schwabach, Germany) for a maximum period of 28 days (series 1) and 14 days (series 2).

Preparation of test dispersions and sampling were performed under aseptic conditions.

### 2.4. Sample Preparation

At each sampling time point, test dispersions were gently swirled and inverted five times to homogenise the dispersion.

For RP-HPLC, pH, and osmolality measurement, 1 mL samples were withdrawn in duplicate on day 0 (immediately after preparation), 1, 3, 5, 7, 14, 21, and 28. Osmolality and pH were measured in undiluted 1 mL samples. For paclitaxel RP-HPLC analysis, 1 mL samples were transferred to 25 mL volumetric flasks. Albumin was precipitated by filling acetonitrile up to the mark of the flasks. After mixing, 5 mL samples were withdrawn and centrifuged for 10 min at 3000 rpm with a Labofuge 400R centrifuge (Heraeus Holding GmbH, Hanau, Germany). For RP-HPLC analysis, 1 mL aliquots of the supernatant were used.

For SEC and DLS analyses, 0.5 mL samples were withdrawn daily from d0 to d14. For SEC analysis, 200 µL aliquots were mixed with 800 µL 0.9% NaCl solution. For DLS analysis, 50 µL aliquots were mixed with 950 µL 0.9% NaCl solution (250 µg/mL paclitaxel).

### 2.5. Visual Inspection

All test dispersions underwent visual inspection for visible particles and colour changes whenever samples were withdrawn. The reconstituted dispersion should be milky and without visible precipitates. If precipitates or settling were visible, the vial was gently inverted to ensure complete redispersion. When complete redispersion was achieved, test dispersions were not considered as out of specification (OOS). If redispersion was not achieved and particulate matter was still observed after the redispersion procedure, test dispersions were considered as OOS.

### 2.6. Particle Size Analysis via DLS

Dynamic light scattering was used to determine the average hydrodynamic diameter (Z-Average) of particles, the polydispersity index (PDI), and the particle size distribution (intensity-based) of the nab-paclitaxel dispersions. Settings of DLS experiments were as follows: UV-cuvette micro, 12.5 × 12.5 × 45 mm (Brand GmbH + Co. KG, Wertheim, Germany); sample refractive index: 1.590; dispersant refractive index: 1.332; viscosity: 0.9207 cP; system temperature: 25.0 ± 0.1 °C; equilibration time: 180 s; run duration: 5 s.

Measurement of particle size distribution by DLS required dilution of nab-paclitaxel test dispersions. Milky, concentrated dispersions cause inaccurate results. Dilution can affect the integrity of the nanoparticles. To determine the optimum measurement concentration and to determine the concentration not causing nanoparticle dissociation, nab-paclitaxel samples of four different concentrations (250 µg/mL, 100 µg/mL, 50 µg/mL, 20 µg/mL) diluted in 0.9% NaCl were prepared and the particle size distribution determined.

The suitability of the method was tested by forced degradation analysis. A total of four 0.2 mL nab-paclitaxel dilutions were prepared with 0.9% NaCl solution. Acidic and alkaline degradation were performed by adding 2 mL 0.1 M hydrochloric acid and 2 mL 0.1 M sodium hydroxide, respectively. Oxidative degradation was performed by adding 2 mL 3% hydrogen peroxide. The samples were heated for 10 min at 70 °C, neutralised (acidic, alkaline samples), and analysed by DLS.

DLS analysis was performed in triplicate (100 runs each) on predetermined time points using the measurement system Zetasizer Nano-ZS (Malvern Instruments Ltd., Malvern, UK). Z-Average and PDI were registered, and size percentiles D10, D50, and D90 were calculated.

### 2.7. Paclitaxel RP-HPLC Assay

Paclitaxel concentrations were determined using a stability indicating RP-HPLC method adapted from Bernabeu et al. [12]. Chromatographic conditions were as follows: column: Nucleodur 100 C18, 5 µm, 150 × 4.6 mm with guard column Universal RP 4 × 3 mm (Macherey-Nagel, Düren, Germany); injection volume: 10 µL; flow rate: 1.3 mL/min; run time: 10 min; sample and column temperature: 20 °C; detection wavelength: 227 nm; mobile phase: 50% acetonitrile + 50% water; pump mode: isocratic.

The RP-HPLC assay was performed using a Waters Alliance 2695 pump connected to a Waters photodiode array detector 2990 (Waters, Eschborn, Germany). Data were collected and processed using Waters Empower 2 software, version 6.10.01.00. Each sample was injected in triplicate by an autosampler.

#### Validation of the RP-HPLC Assay

The RP-HPLC method was validated corresponding to the ICH guideline Q2 (R1) Validation of analytical procedures [13]. Suitability of the method was proven by forced degradation [14]. Aliquots of 0.5 mL Paclitaxel CRS EDQM 500 µg/mL standard solution in acetonitrile were degraded by adding either 0.5 mL of 0.1 M hydrochloric acid or 0.5 mL 0.1 M sodium hydroxide. Oxidative degradation was performed by adding 0.5 mL 3% hydrogen peroxide. Samples were heated for 60 min at 80 °C, neutralised (acidic, alkaline samples), and assayed.

Intra- and inter-day precision and accuracy were tested by assaying ten paclitaxel solutions on three days. A paclitaxel stock solution was freshly prepared each day by dissolving 10 mg paclitaxel CRS in 20 mL acetonitrile. Ten paclitaxel 200 µg/mL solutions were prepared by mixing 400 µL of the paclitaxel stock solution with 600 µL acetonitrile. The first and tenth solution were injected tenfold, solution 2–9 onefold on day 1–3.

Linearity of the RP-HPLC method was tested with a stock solution of 10 mg paclitaxel dissolved in a volumetric flask ad 20 mL with acetonitrile. By further dilution with acetonitrile, solutions of the concentrations 100 µg/mL, 160 µg/mL, 180 µg/mL, 200 µg/mL, 220 µg/mL, 240 µg/mL, and 300 µg/mL were prepared as calibration standards and assayed threefold. The calibration curve was attained by plotting the peak area versus the nominal paclitaxel concentrations.

Precision and accuracy of sample preparation were tested by preparing 6 nab-paclitaxel samples as given above (nominal concentration 200 µg/mL paclitaxel = 100%) which were assayed threefold. To test accuracy within the concentration range of 80–120%, nab-paclitaxel samples with 80% (160 µg/mL) and 120% (240 µg/mL) of the measurement concentration were prepared and assayed threefold.

### 2.8. SEC Analysis (Oligomeric Status of Albumin)

The percentage of albumin monomer and soluble oligomers was analysed by using a size exclusion chromatography method based on the Ph.Eur. monograph 0255 human albumin solution [11]. Chromatographic conditions were as follows: column: TSKgel G3000SWxl 5 µm, 7.8 × 300 mm (Tosoh Bioscience, Griesheim, Germany); injection volume: 10 µL; run time: 20 min; sample temperature: 5 °C; column temperature: 20 °C; detection wavelength: 280 nm; mobile phase: 40 mM phosphate buffer + 11.7 g NaCl in 1000 mL water; pump mode: isocratic.

The SEC assay was performed using a Waters Alliance 2695 pump connected to a Waters photodiode array detector 2990 (Waters, Eschborn, Germany). Data were collected and processed using Waters Empower 2 software, version 6.10.01.00. Each sample was injected in triplicate by an autosampler. Albumin related peaks were quantified by calculating peak areas of each single peak as percentage of the total peak area.

#### Validation of the SEC Assay

Suitability of the SEC assay was tested by forced degradation analysis. Nab-paclitaxel 200 µg/mL dispersions were forced degraded as described in 2.6. and assayed. Intra- and inter-day precision were tested by assaying ten human serum albumin 0.5 mg/mL solutions on three days. Albumin stock solution was prepared by dissolving 10 mg albumin CRS in a 10 mL volumetric flask with 0.9% NaCl. Ten albumin 500 µg/mL solutions were prepared by mixing 500 µL of the albumin stock solution with 500 µL water. The first and tenth solution were injected tenfold, solution 2–9 onefold on day 1–3. Precision of sample preparation was tested by preparing 6 nab-paclitaxel samples as explained in 2.4. (nominal concentration 1 mg/mL paclitaxel) which were assayed threefold.

### 2.9. pH and Osmolality

pH values were determined using a SevenCompact S210 pH metre equipped with an InLab Micro Pro-ISM electrode (Mettler Toledo, Greifensee, Switzerland). The instrument was calibrated weekly with standard buffer solutions of pH 2.00 (batch 1H028J), pH 4.01 (batch 1G207E), pH 7.00 (batch 1H218A), pH 9.21 (batch 1G083C), and pH 11.00 (batch 1G340H) (Mettler Toledo, Greifensee, Switzerland). Functional testing was performed at each sampling time point with a 7.00 standard buffer solution (batch 1H218A, accepted deviation: ± 0.05). Samples were measured once without further dilution.

Osmolality of samples was measured once on a sample volume of 50 µL using an Osmomat 3000 D (Gonotec GmbH, Berlin, Germany). A two-point calibration was performed weekly with a 300 mOsmol/kg calibration standard (Gonotec GmbH, Berlin, Germany) and water HPLC grade. Functional testing was performed at each sampling time point with water HPLC grade (accepted deviation: ±3 mOsmol/kg).

## 3. Results

### 3.1. Visual Inspection

By visual inspection, precipitates were observed in all test dispersions at different time points. Precipitate which was redispersible by swirling and inverting of the vials and EVA bags was not considered as OOS result. Non-redispersible precipitate/particles (considered as OOS result) were detected starting from day 11 (vials stored at 2–8 °C), day 7 (EVA bags stored at 2–8 °C), and day 5 (EVA bags stored at 25 °C) (see Table 2). In test dispersion stored at room temperature, size and number of visible particles increased over time, and yellow colouration was observed after 13 days of storage.

### 3.2. Particle Size Analysis via DLS

#### 3.2.1. Optimum Measurement Concentration and Suitability of the DLS Method

DLS of nab-paclitaxel 50 µg/mL and 20 µg/mL samples revealed several peaks indicating polydispersity and dissociation of paclitaxel-bound albumin nanoparticles to smaller complexes (see Figure 2(2.3,2.4)). DLS of nab-paclitaxel 250 µg/mL and 100 µg/mL dispersions revealed single monomer peaks, indicating monodispersity (see Figure 2(2.1,2.2)). On account of accurate dilution, nab-paclitaxel 250 µg/mL was chosen as optimum concentration for DLS measurement and determination of Z-Average und PDI.

Particle size distributions of forced degraded samples are shown in Appendix A (Appendix A). All types of forced degradation resulted in increased hydrodynamic diameters, indicating aggregation of the nanoparticles. Whereas 70 °C heat resulted in a mean particle size (Z-Average) of about 264 nm, acidic, basic, and oxidative degradation caused substantially increased mean particle sizes of 3.5 µm, 1 µm, and 4.5 µm, respectively.

#### 3.2.2. Particle Size Analysis of Reconstituted Nab-Paclitaxel 5 mg/mL Dispersions Stored in Original Vials and EVA Infusion Bags

The Z-Average, PDI, and particle size distribution of test dispersions met the specifications (see Table 1) until day 11 when stored refrigerated (vials and EVA infusion bags at 2–8 °C) and until day 6 when stored in EVA infusion bags at 25 °C. Z-Average increased moderately and PDI doubled over the 14-day observation period when test dispersions were stored refrigerated (see Table 3 and Table 4). In contrast, Z-Average increased significantly and PDI tripled over the 14-day observation period when test dispersions in EVA bags were stored at 25 °C. Starting at day 8 of storage, Z-Average strongly increased, indicating temperature-dependent aggregation (see Table 5).

### 3.3. Paclitaxel RP-HPLC Assay

#### 3.3.1. Validation of the RP-HPLC Assay


*Suitability*


Chromatograms of forced degraded paclitaxel CRS solutions are shown in Appendix A (Appendix A). The paclitaxel parent peak elutes with a retention time (Rt) of about 4.5 min. Chromatograms of heated paclitaxel solutions showed only the parent peak and no peaks of degradation products. Application of hydrochloric acid combined with heat resulted in several peaks of degradation products with Rt < 4.5 and one peak with Rt 7.8 min. After base plus heat induced degradation, the paclitaxel parent peak was not detectable any longer and several peaks of degradation products with very short Rts were registered. Peaks are related to hydrolysis of paclitaxel under alkaline and acidic conditions. Oxidative degradation resulted in a distinctly reduced peak area of the paclitaxel parent peak and peaks of degradation products at Rt 0.5 and 1 min. No secondary peak interfered with the paclitaxel parent peak.


*Intra- and inter-day precision, accuracy, and linearity*


The intra-day and inter-day precision assays revealed a mean paclitaxel concentration of 200.02 µg/mL ± 0.35% relative standard deviation (RSD), and 204.48 µg/mL ± 1.68% RSD, respectively (acceptance criteria RSD < 2%). Accuracy was determined via recovery rate of the mean measured paclitaxel concentration throughout the validation period and amounted to 102.24% (acceptance criteria 95–105%). The coefficient of correlation attained by plotting the peak areas against the nominal paclitaxel concentrations was R^2^ = 0.9999 (acceptance criteria R^2^ > 0.99).

Precision of sample preparation revealed a mean paclitaxel concentration of 200.09 µg/mL ± 1.29% RSD and an accuracy of 100.4% for samples with 100% nominal concentration. Accuracy of samples with 80% and 120% nominal concentration was 101.08% and 101.30%, respectively.

#### 3.3.2. Paclitaxel Concentration in Reconstituted Nab-Paclitaxel 5 mg/mL Dispersions Stored in Original Vials and EVA Infusion Bags

Paclitaxel concentrations of reconstituted nab-paclitaxel dispersions stored refrigerated (2–8 °C) in the original vials and in EVA infusion bags remained nearly unchanged (concentration decrease 2–3%) over a period of 7 days and 14 days, respectively (Table 6). Test dispersions stored in EVA infusion bags at 25 °C also remained stable up to day 7 (>95% of the initial concentration) and subsequently declined rapidly (Table 6). Afterwards, paclitaxel concentrations declined far below the acceptance level. In parallel, non-redispersible particles became obvious. In none of the HPLC chromatograms secondary peaks of paclitaxel degradation products were detected.

### 3.4. SEC Analysis (Oligomeric Status of Albumin)

#### 3.4.1. Validation of the SEC Assay

In SEC chromatograms of reconstituted nab-paclitaxel test dispersions (see Figure 3) four peaks related to albumin were present and identified according to the relative retention times (rRt):

1. albumin polymer (rRt 0.67)

2. oligomer 1 (rRt 0.77)

3. dimer (rRt 0.88)

4. albumin monomer (rRt 1).

In addition, a peak related to N-acetyl-DL-tryptophan (rRt 1.95) became obvious in the chromatograms. The substance is used as excipient in the licensed medicinal product.


*Suitability*


Chromatograms of forced degraded reconstituted nab-paclitaxel dispersions (Pazenir 5 mg/mL) are shown in Appendix A (Appendix A). Heating of nab-paclitaxel test dispersions diluted with 0.9% NaCl solution resulted in a decrease of peak areas of polymers and dimers. Under acidic conditions a decrease of the polymer related peak area was found. Alkaline and oxidative conditions caused significant albumin degradation and distinct decreases of the monomer peak. Additional peaks appeared, which were interpreted as albumin fragments.


*Intra- and inter-day precision*


The intra-day and inter-day precision assay revealed a mean albumin monomer relative peak area of 96.28% ± 0.58% RSD and 96.23% ± 0.52% RSD, respectively. Precision of sample preparation revealed a mean albumin monomer relative peak area of 85.63% ± 0.46% RSD. RSD values of precision assays < 5% were accepted.

#### 3.4.2. Oligomeric Status of Albumin in Reconstituted Nab-Paclitaxel 5 mg/mL Dispersions Stored in Original Vials and EVA Infusion Bags

Over the 14-day observation period, the total peak area of albumin species decreased about 4% compared to the initial total peak area independent from the type of primary container or storage temperature (Table 7). All test dispersions met the specifications (≥85% albumin monomer, ≤10% albumin oligomer 1 plus polymer) when assayed daily, from day 0 to day 14. The percentage of the monomer peak (in relation to the albumin total peak area) increased in all test dispersions over time (Table 8). Relative peak areas of the albumin polymer peak decreased over time, predominantly in dispersions stored at 25 °C.

### 3.5. pH and Osmolality

Measured pH values of nab-paclitaxel test dispersions decreased slightly over time but remained within the predefined limits (pH 6.85–7.5). Osmolality remained unchanged in all test dispersions over time and within the accepted range of 335–345 mOsmol/kg.

## 4. Discussion

Various instability reactions of nab-paclitaxel preparations like disintegration, aggregation of the nanoparticles [4], release and precipitation of paclitaxel [8,9,10], and degradation reactions of albumin [11] affect each other. Common external factors affecting its in-use stability are storage temperature and the material of the primary container, whereas different concentrations are not relevant for stability studies, since nab-paclitaxel dispersions are administered undiluted. According to the complexity of the degradation reactions and in-line with relevant guidelines [15,16], orthogonal analytical methods like DLS, SEC, RP-HPLC, and visual inspection were utilised to investigate the physicochemical stability of RTU and RTA Pazenir preparations. Specifications were set as given in the manufacturer’s certificate of analysis [17]. Of note, batch variability did not become obvious when three different batches were analysed. Knowing the growth promoting nature of nab-paclitaxel [18], special attention was paid to the aseptic preparation and sampling of test dispersions to avoid microbiological contamination and microbiological instability reactions.


*Integrity of nab-paclitaxel nanoparticles*


Aggregation of the nab-paclitaxel nanoparticles and release of paclitaxel proved to be the stability-determining factor. As given in the SmPC, redispersible precipitate occurred mainly in glass vials and was not considered as out of specification. The occurrence of non-redispersible precipitate/particles, which indicate a potential loss of efficacy and safety, was dependent on external factors. Test dispersions stored in EVA bags were more likely to show precipitation than test dispersions stored in glass vials. Trissel supposed the configuration of the interior container surface as a stimulus for paclitaxel precipitation [19]. The increased and rougher surface area of 150 mL EVA bags probably favoured precipitation compared to 50 mL glass vials. Further, paclitaxel precipitation occurred distinctly earlier and to greater extent in test dispersions stored at room temperature, since higher temperature favours the release of paclitaxel from albumin.

Non-redispersible particles were detected by visual inspection at sampling time points, when results of DLS analysis still fulfilled the specifications set (see Table 1). This can be explained by the operational range of DLS analysis (particle size of about 1 nm–10 µm), whereby particle sizes beyond 10 µm cannot be determined precisely. In contrast, visual inspection detects particles > 100 µm. The average hydrodynamic diameter (Z-Average) and PDI increased continuously over time, predominantly in test dispersions stored at 25 °C, confirming that higher temperature causes instability of nab-paclitaxel dispersions. The PDI proved to be the most sensitive parameter of DLS analysis. Monodispersity (PDI ≤ 0.20) was given until day 11 in all test dispersions stored refrigerated, independent from the container material. At room temperature, this limit was met only until day 6.

Particle size distribution plots of DLS analysis did not indicate any disintegration of the nanoparticles. These results are reliable as the suitability of the measurement concentration (250 µg/mL) was proven upfront.

As Terkola et al. [7] reported only the results of DLS analysis of nab-paclitaxel dispersions in terms of Z-average and PDI without limits set, comparison with the results of our study is not advisable.


*Integrity of paclitaxel*


In-use stability of paclitaxel infusion solutions formulated with Cremophor solubilizer is described as concentration and temperature dependent [8,9,10]. Precipitation is known as a stability limiting factor. Stability of nab-paclitaxel infusion dispersions was also limited by precipitation of released paclitaxel. Paclitaxel concentrations in test dispersions decreased distinctly after 7 days of storage, but no signs of chemical degradation were found in the HPLC chromatograms. Moreover, abnormal RSD values of the measured paclitaxel concentrations and non-redispersible particles were detected in parallel with the decrease of paclitaxel concentrations, indicating physical instability and inhomogeneous test dispersions.


*Integrity of albumin*


Protein instability can occur in terms of denaturation, fragmentation, aggregation, un- or misfolding, or chemical degradation to the amino acid residues. Instability can be caused by external factors as temperature, light exposure, pH shifts, exposure to oxygen, or shear stress [20]. Human serum albumin is a highly soluble and robust protein. Albumin-containing medicinal products are generally stabilised with sodium caprylate and N-acetyl-DL-tryptophan [21] and pasteurised at 60 °C for 10 h without impact on structural integrity [11]. SEC was used to determine albumin fragmentation and the formation of albumin oligomers and soluble polymers, which could impair nanoparticle stability.

During the observation period, the total peak area of soluble albumin decreased by a maximum of 5%. In parallel, the relative peak area of soluble polymers decreased, shifting the albumin species ratio towards the albumin monomer. These findings can be explained by the formation of insoluble albumin aggregates not detectable by SEC.

The occurrence of visible, non-redispersible particles was not predictable, and time points when visible particles appeared first differed within the sample triplicates. Taking into account DLS and HPLC results, RTA dispersions stored at 2–8 °C were considered to be stable for 7 days, provided that visual inspection of the dispersions does not reveal any non-redispersible precipitate.


*Limitations*


In our study, visual inspection was an important tool to detect instability of nab-paclitaxel dispersions because the operational range of DLS was not suitable to detect the precipitate. However, visual inspection is generally impaired by subjectivity. In addition, the milky appearance of the test dispersions hindered visual inspection. Similarly, measurement of sub-visible particles with the particle counter (Ph.Eur. 2.9.19 method 1) was not possible because of the milky appearance of the dispersion. The more suitable method, membrane microscopy (Ph.Eur. 2.9.19 method 2), is not implemented in our laboratory.

Zeta potential represents another parameter to predict aggregation of nanoparticles. As the measurement of zeta potential and particle size distribution in parallel in an identical sample was not feasible, we preferred particle size measurement.

Because of time-constraints, it was not possible to apply each analytical method in parallel on the samples taken at a specified time point. Therefore, test dispersions were prepared and analysed in two series. It was initially planned to use the same pattern of sampling time points (day 0, 1, 3, 5, 7, 14, 21, 28) in the first and second series. After non-redispersible particles have already been observed on day 5 of storage in the first series, the sampling plan was accordingly adapted in the second series. Visual inspection, DLS, SEC were conducted daily over a maximum period of 14 days.

## 5. Conclusions

Stability of nab-paclitaxel dispersions is limited by the release and crystallisation of water-insoluble paclitaxel from the nanoparticles and formation of insoluble albumin aggregates, indicated by non-redispersible visible particles. Based on the overall study results, the shelf life of reconstituted Pazenir 5 mg/mL dispersions stored at 2–8 °C either in original glass vials (RTU) or in EVA infusion bags (RTA) is limited to 7 days, provided that any precipitate is redispersible. The shelf life of Pazenir 5 mg/mL RTA preparations stored at 25 °C in EVA infusion bags is limited to 4 days, provided that any precipitate is redispersible. Careful visual inspection of nab-paclitaxel dispersions after reconstitution and prior to administration is highly recommended.

Since physicochemical stability of reconstituted nab-paclitaxel dispersions appeared to be temperature dependent, refrigerated storage is recommended. To prevent microbial contamination, reconstitution and preparation of Pazenir RTA-dispersions should be performed under controlled aseptic conditions.

## Figures and Tables

**Figure 1 pharmaceutics-16-01372-f001:**
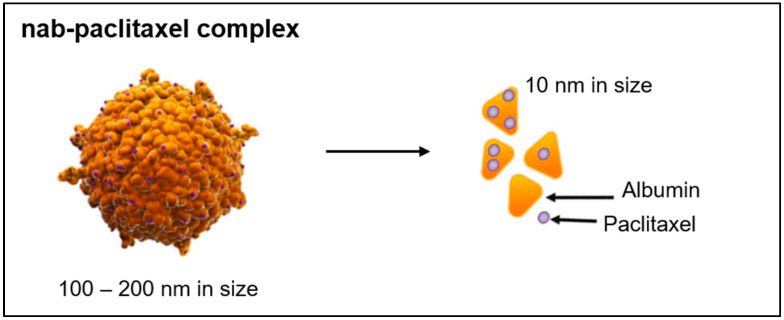
Scheme of nab-paclitaxel complex adopted from [5].

**Figure 2 pharmaceutics-16-01372-f002:**
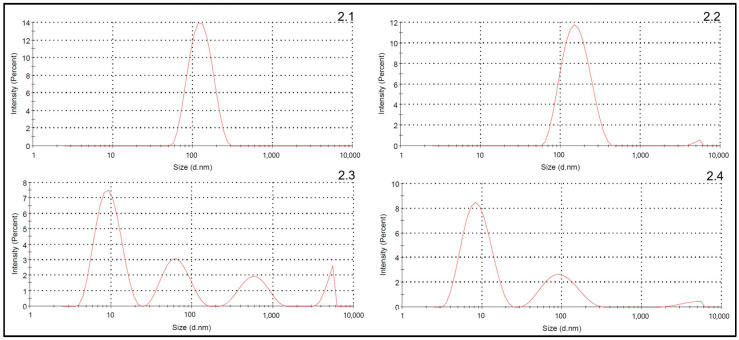
Particle size distribution of nab-paclitaxel 250 µg/mL (**2.1**), 100 µg/mL (**2.2**), 50 µg/mL (**2.3**) and 20 µg/mL (**2.4**) dilutions analysed by DLS.

**Figure 3 pharmaceutics-16-01372-f003:**
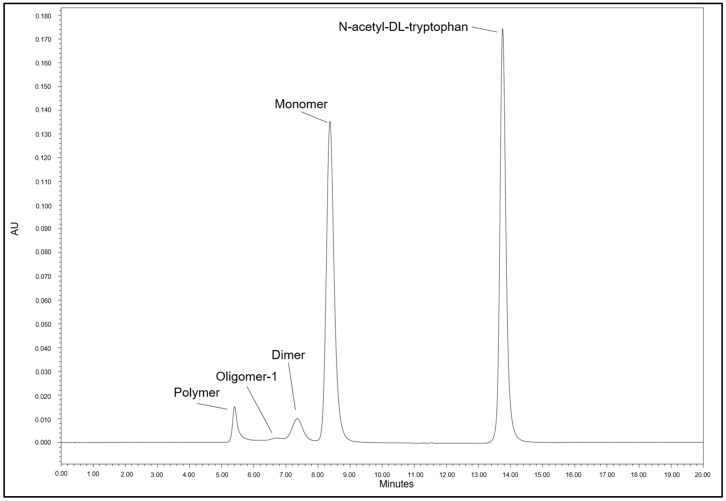
SEC chromatogram of freshly reconstituted nab-paclitaxel (Pazenir 5 mg/mL) dispersion diluted to a concentration of 1 mg/mL nab-paclitaxel.

**Table 1 pharmaceutics-16-01372-t001:** Methods for stability testing and specifications set for nab-paclitaxel dispersions (brand product: Pazenir 5 mg/mL).

Method/Test Parameter.	Specification
Visual inspection	White to slightly yellow, milky, homogenous dispersion without non-redispersible precipitate and particulate matter
Osmolality Ph.Eur. 2.2.35	300–360 mOsmol/kg
pH Ph.Eur. 2.2.3	6.0–7.5
DLS/Particle size	
● Average hydrodynamic diameter (Z-Average)	95–160 nm
● Polydispersity index (PDI)	≤0.20
● Size percentile D10	≥74 nm
● Size percentile D50	95–160 nm
● Size percentile D90	≤218 nm
SEC Ph.Eur. 2.2.30/Oligomeric status of Albumin	
Albumin Monomer	≥85%
Albumin oligomer 1 plus albumin polymer	≤10%
RP-HPLC Ph.Eur. 2.2.29/Paclitaxel concentration	95.0–105.0%

**Table 2 pharmaceutics-16-01372-t002:** Results of visual examination for visible particles and colour changes in reconstituted nab-paclitaxel dispersions (Pazenir 5 mg/mL) stored in the original glass vials or EVA infusion bags under different temperature conditions over 14 days.

Storage Condition	Particle Formation/Colour Change	d 0	d 1	d 2	d 3	d 4	d 5	d 6	d 7	d 8	d 9	d 10	d 11	d 12	d 13	d 14
Original vial 2–8 °C	redispersible precipitate	-	-	-	-	-	-	^✔^(2)	^✔^(2)	^✔^(2)	^✔^(2)	^✔^(2)	-	-	-	-
	non-redispersible precipitate/particles	-	-	-	-	-	-	-	-	-	-	-	^✔^(3)	^✔^(3)	^✔^(3)	^✔^(3)
	yellow colouration	-	-	-	-	-	-	-	-	-	-	-	-	-	-	-
EVA infusion bag 2–8 °C	redispersible precipitate	-	-	-	-	-	-	-	-	^✔^(2)	-	-	-	-	-	-
	non-redispersible precipitate/particles	-	-	-	-	-	-	-	^✔^(1)	^✔^(1)	^✔^(3)	^✔^(3)	^✔^(3)	^✔^(3)	^✔^(3)	^✔^(3)
	yellow colouration	-	-	-	-	-	-	-	-	-	-	-	-	-	-	-
EVA infusion bag 25 °C	redispersible precipitate	-	-	-	-	-	-	-	-	-	-	-	-	-	-	-
	non-redispersible precipitate/particles	-	-	-	-	-	^✔^(2)	^✔^(3)	^✔^(3)	^✔^(3)	^✔^(3)	^✔^(3)	^✔^(3)	^✔^(3)	^✔^(3)	^✔^(3)
	yellow colouration	-	-	-	-	-	-	-	-	-	-	-	-	-	^✔^(3)	^✔^(3)

^✔^(1) = particles/yellow colouration in one of three test dispersions, (2) = in two of three test dispersions, (3) = in all three test dispersions; - = particles or colouration were not observed.

**Table 3 pharmaceutics-16-01372-t003:** Intensity-based mean particle diameter (Z-Average), polydispersity index (PDI) and D10, D50 and D90 of reconstituted nab-paclitaxel dispersions (Pazenir 5 mg/mL) stored in the original glass vials at 2–8 °C for 14 days (n = 3). OOS-results are shaded in grey.

Day of Storage	Z-Average [nm] ± RSD [%]	PDI ± RSD [%]	D10 [nm] ± RSD [%]	D50 [nm] ± RSD [%]	D90 [nm] ± RSD [%]
d 0	117 ± 0.67	0.10 ± 0.70	81 ± 2.46	124 ± 0.57	190 ± 0.74
d 1	120 ± 2.66	0.13 ± 38.86	81 ± 4.39	127 ± 2.41	203 ± 10.27
d 2	120 ± 3.36	0.12 ± 30.64	81 ± 2.92	128 ± 4.13	204 ± 11.26
d 3	118 ± 0.90	0.11 ± 6.67	81 ± 1.57	125 ± 0.80	194 ± 1.86
d 4	119 ± 1.67	0.10 ± 4.58	83 ± 2.77	126 ± 1.59	193 ± 0.79
d 5	120 ± 1.75	0.10 ± 16.05	84 ± 0.62	128 ± 2.39	195 ± 6.19
d 6	120 ± 1.35	0.11 ± 5.99	83 ± 1.07	128 ± 1.35	200 ± 3.69
d 7	122 ± 2.17	0.11 ± 15.92	84 ± 2.62	130 ± 2.23	203 ± 6.42
d 8	125 ± 3.17	0.12 ± 26.94	84 ± 2.39	132 ± 3.47	210 ± 9.57
d 9	126 ± 3.15	0.13 ± 23.79	85 ± 1.12	133 ± 1.50	210 ± 2.63
d 10	127 ± 3.55	0.13 ± 25.42	86 ± 2.80	133 ± 1.99	209 ± 5.02
d 11	130 ± 5.10	0.16 ± 30.86	86 ± 2.66	136 ± 1.27	218 ± 2.53
d 12	136 ± 7.23	0.21 ± 31.07	88 ± 5.55	134 ± 1.72	208 ± 3.10
d 13	137 ± 5.47	0.20 ± 28.76	88 ± 3.53	138 ± 2.22	222 ± 0.94
d 14	145 ± 9.37	0.22 ± 34.81	92 ± 5.61	140 ± 1.80	215 ± 6.20

**Table 4 pharmaceutics-16-01372-t004:** Intensity-based mean particle diameter (Z-Average), polydispersity index (PDI) and D10, D50 and D90 of reconstituted nab-paclitaxel dispersions (Pazenir 5 mg/mL) stored in EVA infusion bags at 2–8 °C for 14 days (n = 3). OOS-results are shaded in grey.

Day of Storage	Z-Average [nm] ± RSD [%]	PDI ± RSD [%]	D10 [nm] ± RSD [%]	D50 [nm] ± RSD [%]	D90 [nm] ± RSD [%]
d 0	116 ± 0.78	0.09 ± 8.98	81 ± 1.30	123 ± 1.25	186 ± 3.36
d 1	119 ± 3.59	0.12 ± 30.28	81 ± 2.87	126 ± 3.46	199 ± 10.73
d 2	120 ± 4.13	0.13 ± 38.69	81 ± 2.87	127 ± 3.43	200 ± 9.85
d 3	118 ± 1.46	0.1 ± 5.16	83 ± 1.17	124 ± 1.67	188 ± 2.32
d 4	118 ± 1.03	0.1 ± 7.00	82 ± 2.87	125 ± 0.80	191 ± 1.89
d 5	123 ± 5.45	0.13 ± 35.89	81 ± 1.37	131 ± 5.54	213 ± 11.95
d 6	119 ± 1.40	0.1 ± 10.91	84 ± 1.55	126 ± 1.83	190 ± 3.73
d 7	121 ± 2.08	0.11 ± 3.53	84 ± 1.58	129 ± 2.05	199 ± 2.94
d 8	124 ± 2.68	0.11 ± 27.58	84 ± 1.84	130 ± 2.69	205 ± 8.72
d 9	125 ± 4.44	0.13 ± 28.05	84 ± 1.42	132 ± 4.32	209 ± 6.80
d 10	128 ± 5.61	0.17 ± 36.64	85 ± 3.07	131 ± 3.17	208 ± 5.37
d 11	132 ± 8.30	0.19 ± 39.67	85 ± 3.45	133 ± 3.38	214 ± 4.35
d 12	136 ± 10.53	0.21 ± 50.18	87 ± 2.91	135 ± 4.07	216 ± 7.18
d 13	157 ± 20.54	0.27 ± 53.10	90 ± 3.40	141 ± 4.29	239 ± 21.74
d 14	177 ± 34.67	0.28 ± 50.43	92 ± 5.21	141 ± 4.26	323 ± 57.39

**Table 5 pharmaceutics-16-01372-t005:** Intensity-based mean particle diameter (Z-Average), polydispersity index (PDI) and D10, D50 and D90 of reconstituted nab-paclitaxel dispersions (Pazenir 5 mg/mL) stored in EVA infusion bags at 25 °C for 14 days (n = 3). OOS-results are shaded in grey.

Day of Storage	Z-Average [nm] ± RSD [%]	PDI ± RSD [%]	D10 [nm] ± RSD [%]	D50 [nm] ± RSD [%]	D90 [nm] ± RSD [%]
d 0	131 ± n.a.	0.2 ± n.a.	82 ± n.a.	135 ± n.a.	214 ± n.a.
d 1	118 ± 1.27	0.1 ± 2.60	83 ± 1.46	125 ± 1.23	187 ± 1.11
d 2	121 ± 1.48	0.11 ± 26.38	83 ± 2.85	128 ± 1.96	200 ± 7.80
d 3	121 ± 1.58	0.11 ± 4.68	84 ± 1.24	129 ± 1.18	201 ± 3.31
d 4	124 ± 1.99	0.13 ± 14.76	83 ± 1.50	132 ± 2.27	212 ± 6.99
d 5	127 ± 3.53	0.13 ± 34.58	86 ± 2.18	134 ± 2.40	208 ± 6.89
d 6	130 ± 5.34	0.17 ± 35.90	85 ± 1.50	134 ± 1.97	216 ± 5.26
d 7	137 ± 8.27	0.22 ± 36.26	86 ± 1.96	136 ± 1.13	223 ± 3.83
d 8	159 ± 23.48	0.3 ± 43.78	88 ± 3.08	135 ± 0.85	348 ± 63.61
d 9	236 ± 65.58	0.38 ± 41.37	93 ± 6.07	140 ± 2.29	699 ± 109.35
d 10	1930 ± 149.06	0.39 ± 28.69	972 ± 155.85	1192 ± 151.70	1910 ± 100.89
d 11	2633 ± 99.69	0.57 ± 66.71	845 ± 150.16	1770 ± 81.97	2983 ± 41.90
d 12	3110 ± 51.15	0.54 ± 66.94	1437 ± 81.33	3143 ± 9.41	4803 ± 15.99
d 13	3451 ± 19.94	0.29 ± 17.70	2237 ± 33.09	3000 ± 28.92	4217 ± 27.88
d 14	2888 ± 12.94	0.32 ± 3.04	1770 ± 8.83	2430 ± 15.23	3440 ± 25.05

**Table 6 pharmaceutics-16-01372-t006:** Paclitaxel concentrations of reconstituted nab-paclitaxel dispersions (Pazenir 5 mg/mL) stored in the original glass vials or EVA infusion bags under different temperature conditions for 28 days. OOS-results are shaded in grey.

Storage Condition	Initial Paclitaxel Concentration [mg/mL] ± RSD [%]		Percentage Rate of the Initial Paclitaxel Concentration ± RSD [%] (Concentration at d 0 = 100%) (n = 9)						
	Nominal	Measured (d 0)	d 1	d 3	d 5	d 7	d 14	d 21	d 28
Original vial 2–8 °C	5	5.23 ± 1.1	97.85 ± 1.1	97.37 ± 1.1	98.19 ± 0.8	97.69 ± 0.4	83.91 ± 9.6	64.81 ± 20.0	38.33 ± 37.1
EVA infusion bag 2–8 °C	5	5.15 ± 2.2	100.5 ± 2.1	98.35 ± 1.1	99.07 ± 1.8	98.39 ± 1.7	97.05 ± 2.1	59.48 ± 52.7	67.87 ± 36.1
EVA infusion bag 25 °C	5	5.17 ± 3.0	98.66 ± 2.3	97.92 ± 1.2	97.22 ± 1.6	97.77 ± 1.6	71.74 ± 9.7	57.64 ± 10.8	60.06 ± 12.1

**Table 7 pharmaceutics-16-01372-t007:** Total albumin peak area determined by SEC in Pazenir dispersions stored under different conditions for 14 days.

Storage Condition	Total Peak Area of Albumin in Relation to the Initial Total Peak Area (d 0 = 100%) (n = 9)														
	d 0	d 1	d 2	d 3	d 4	d 5	d 6	d 7	d 8	d 9	d 10	d 11	d 12	d 13	d 14
Original vial 2–8 °C	100	99.99	100.21	100.77	99.99	100.44	100.09	98.96	100.01	99.34	99.78	98.61	97.20	97.60	94.99
EVA infusion bag 2–8 °C	100	99.65	99.95	100.39	100.39	101.00	100.91	100.05	101.10	99.36	99.92	99.07	98.06	97.67	95.96
EVA infusion bag 25 °C	100	99.99	100.46	100.56	100.23	100.12	101.08	100.54	99.58	97.47	97.14	96.88	95.46	98.09	96.59

**Table 8 pharmaceutics-16-01372-t008:** Peak areas of albumin polymer, oligomer 1, dimer, and monomer expressed as percentage rate of the total peak area determined by SEC in reconstituted nab-paclitaxel dispersions (Pazenir 5 mg/mL) stored in the original glass vials or EVA infusion bags under different temperature conditions over 14 days.

Storage Condition	Albumin Species	Peak Areas [%] ± Relative Standard Deviation (RSD) [%] of Albumin Species in Relation to the Albumin Total Peak Area (n = 9)				
		d 0	d 3	d 7	d 10	d 14
Original vial 2–8 °C	Polymer	6.23 ± 1.64	6.33 ± 0.99	6.12 ± 3.06	5.48 ± 9.07	3.34 ± 30.41
	Oligomer-1	0.64 ± 7.33	0.61 ± 10.48	0.53 ± 15.23	0.51 ± 15.91	0.57 ± 9.71
	Dimer	7.9 ± 0.67	7.85 ± 1.24	7.76 ± 1.3	7.77 ± 2.18	7.88 ± 1.95
	Monomer	85.22 ± 0.19	85.21 ± 0.19	85.59 ± 0.17	86.24 ± 0.5	88.22 ± 0.95
EVA infusion bag 2–8 °C	Polymer	6.29 ± 1.02	6.35 ± 1.01	6.17 ± 2.37	5.59 ± 9.89	2.5 ± 77.46
	Oligomer-1	0.59 ± 12.95	0.57 ± 10.92	0.56 ± 9.36	0.53 ± 10.71	0.5 ± 9.33
	Dimer	7.9 ± 1.28	7.8 ± 0.84	7.73 ± 0.88	7.75 ± 1.4	7.83 ± 1.93
	Monomer	85.21 ± 0.2	85.28 ± 0.15	85.53 ± 0.17	86.14 ± 0.54	89.16 ± 2.04
EVA infusion bag 25 °C	Polymer	6.28 ± 0.72	6.23 ± 2.31	4.69 ± 18.17	2.3 ± 47.14	1.96 ± 4.21
	Oligomer-1	0.62 ± 9.66	0.55 ± 9.43	0.54 ± 9.71	0.5 ± 9.58	0.56 ± 14.74
	Dimer	7.88 ± 1.27	7.62 ± 1.29	7.68 ± 1.38	7.78 ± 1.26	7.82 ± 1.93
	Monomer	85.22 ± 0.18	85.6 ± 0.27	87.09 ± 0.97	89.42 ± 1.15	89.66 ± 0.32

## Data Availability

Data are contained within the article and Appendix A.

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
