# Peer review of "Physicochemical Stability of Nab-Paclitaxel (Pazenir) Infusion Dispersions in Original Glass Vials and EVA Infusion Bags"

_pharmaceutics, 2024, doi:10.3390/pharmaceutics16111372_

Round 1
Reviewer 1 Report
Comments and Suggestions for Authors
The present study effectively demonstrates the stability of nab-paclitaxel formulations, and these findings are anticipated to be valuable in informing the storage and administration practices of the formulation in clinical settings. However, several revisions should further improve the completeness and quality of this paper.
1. Lack of Smooth Transitions in the Introduction
The introduction is well-structured but jumps between concepts without smooth transitions, especially when discussing nab-technology, its benefits, and the specific study objectives. A transition sentence could be added when moving from the description of nab-technology to the specifics of the drug Pazenir. For example, after the sentence ending with "making premedication unnecessary," you could add, "Given these advantages, Pazenir has emerged as a significant alternative to traditional formulations of paclitaxel."
2. Clarification Needed for "Non-Redispersible Particles"
Clarify the term "non-redispersible particles" by briefly explaining why their formation is problematic. For example, "Non-redispersible particles, which indicate a loss of homogeneity and potential efficacy of the dispersion, were observed..."
3. Conclusion for Introduction Lacking
The introduction ends somewhat abruptly after listing the methods. Add a concluding sentence that emphasizes the importance of the study’s findings in the broader context of pharmaceutical stability research. For example, "This study provides essential insights into the stability of nab-paclitaxel formulations, contributing to better storage and administration practices in clinical settings."
4. Ambiguity in Criteria for Visual Inspection
The criteria for "redispersible precipitate" and "non-redispersible particles" are unclear. This ambiguity can affect the interpretation and reproducibility of the results. Clearly define the criteria for visual inspection results and specify what each term means.
5. Lack of Detailed Explanations in Results and Discussion
There are several areas where more detailed explanations are needed in the Results and Discussion sections. Add clarifying sentences for the following points:
1) Reason for Choosing the Nab-Paclitaxel Concentration in DLS Analysis
The reason for selecting the nab-paclitaxel concentration (250 µg/mL) used in DLS analysis is not sufficiently explained. Add a sentence explaining why the selected concentration is more appropriate than others.
2) Comparison of Stability Under Different Temperature Conditions
The comparison of stability differences under different temperature conditions is not clearly articulated. Add sentences that clearly compare and discuss the stability of nab-paclitaxel under various temperature conditions. For instance, "Nab-paclitaxel dispersions became unstable more quickly when stored at 25 °C, suggesting that higher temperatures promote nanoparticle aggregation."
3) Explanation of Peaks in HPLC Results
There is insufficient explanation of what each peak represents and its significance in the HPLC results. Add sentences to explain the meaning of each peak and the importance of peaks that appear under specific conditions.
4) Explanation of Albumin Stability
The explanation of albumin stability is inadequate, particularly regarding the interpretation of changes in the ratio of albumin monomers, dimers, and oligomers. Include a sentence like, "A decrease in the proportion of albumin oligomers suggests a decline in protein stability, which could impact the long-term efficacy of nab-paclitaxel."
6. Lack of Practical Recommendations in Conclusion
The conclusion lacks practical clinical recommendations based on the study findings. Provide specific clinical recommendations based on the study outcomes. Additionally, the description of the study’s limitations is relatively brief. Clearly explain the limitations and discuss how these limitations might have influenced the results. For instance, "Due to equipment limitations in the laboratory, certain analytical methods could not be utilized, which may have impacted the comprehensiveness of the results." Add detailed explanations like this.
Comments on the Quality of English Language
Revisions are needed to enhance readability by improving the flow between sentences.
Reviewer 2 Report
Comments and Suggestions for Authors
In this work, the authors presented the aggregative behavior of nab-paclitaxel ready-to-use dispersion for infusion in original glass vials and ready-to-administer infusion dispersion in EVA infusion bags. Although the significance of novelty may be questioned, the manuscript is prepared very carefully and clearly. The study has an obvious goal and the authors did it meticulously, describing everything in so details that there are almost no questions. Only a slight expansion of the discussions is recommended. For example, are there other pharmaceutical formulations for which the same result (worst stability in EVA bags in comparison with glass vials) has been recorded? The authors also need to clarify why they did not use UV-Vis absorption spectroscopy in their study. Some minor design flaws should also be corrected.
Comments on the Quality of English LanguageMinor editing of English language required.
Reviewer 3 Report
Comments and Suggestions for Authors
In this article, Linxweiler et al. studies the physicochemical stability of an albumin-bound paclitaxel ready-to-use (RTU) dispersion for infusion in original glass vials and ready-to-administer (RTA) infusion dispersion in EVA infusion bags. The authors studied the physicochemical stability through dynamic light scattering, size-exclusion-chromatography and visual inspection. Chemical stability was followed by RP-HPLC, osmolality and pH measurements. The article is well written, presenting some points that need a clarification and revision. The suggestions are as follows:
1) In line 68 and 77, the abbreviations “SmPC” and “EVA” should be defined, respectively.
2) The nanoparticle aggregation is commonly associated with changes in the nanoparticle surface charge. The authors are suggested to carry out zeta-potential measurements of the samples.
3) In line 446, the citation is missing after “Terkola et al.”
4) The correlograms of the DLS data in Figure 2 should be available in Supplementary Materials.
5) The document is not in accordance with the journal guidelines regarding the Supplementary materials. For example, Figure 4 should be Figure S1. Besides, the images should be provided in a word file, in which the respective titles and captions are provided.
6) The authors are suggested to present the standard deviation instead of relative standard deviation to ease the comparison of the values.
7) Did the authors test the forced degradation of only paclitaxel or only BSA? This should be carried out as a control for the SEC chromatogram studies.
Reviewer 4 Report
Comments and Suggestions for Authors
The manuscript submitted by Linxweiler et al. focuses on the characterization of nab-paclitaxel formulations for infusion. The manuscript is very well written and organized. Results are clearly presented; the discussion is comprehensive with limitations emphasized. The conclusions are supported by the results. In my opinion, the manuscript can be accepted after minor revision.
1. Section 2.1. The introductory paragraph with the brief explanations and reference to the Table 1 should be provided (before Table 1).
2. Tables 3-6 and 8. It seems better to present RSD as a value in the same units as the mean value, but not as a percentage. In my opinion, presenting RSD as a % makes the data more difficult to understand.
Reviewer 5 Report
Comments and Suggestions for Authors
The manuscript describes the results of analysis of the stability of nab-paclitaxel (Pazenir) infusion dispersions in original glass vials and EVA infusion bags. Since paclitaxel is widely used in cancer chemotherapy, the study on analysis of ist nanoparticulate-based form may be interesting for medical doctors, scientists and pharmacists. The manuscript is well written, comprehenisive and presents some solutions that may be useful in practice. Therefore, I recommend to accept the manuscript in the present form.
Round 2
Reviewer 3 Report
Comments and Suggestions for Authors
The authors considered all the suggestions. Thus, acceptance of the manuscript is recommended.